# Hypoxia Affects the Antioxidant Activity of Glutaredoxin 3 in *Scylla paramamosain* through Hypoxia Response Elements

**DOI:** 10.3390/antiox12010076

**Published:** 2022-12-29

**Authors:** Yu-Kun Jie, Chang-Hong Cheng, Hong-Ling Ma, Guang-Xin Liu, Si-Gang Fan, Jian-Jun Jiang, Zhi-Xun Guo

**Affiliations:** 1Key Laboratory of South China Sea Fishery Resources Exploitation & Utilization, Ministry of Agriculture and Rural Affairs, South China Sea Fisheries Research Institute, Chinese Academy of Fishery Sciences, Guangzhou 510300, China; 2National Demonstration Center for Experimental Fisheries Science Education, Shanghai Engineering Research Center of Aquaculture, Shanghai Ocean University, Shanghai 201306, China

**Keywords:** hypoxia, glutaredoxin, HIF-1, HRE, *Scylla paramamosain*

## Abstract

Hypoxia is a major environmental stressor that can damage the oxidation metabolism of crustaceans. Glutaredoxin (Grx) is a key member of the thioredoxin superfamily and plays an important role in the host’s defense against oxidative stress. At present, the role of Grx in response to hypoxia in crustaceans remains unclear. In this study, the full-length cDNA of Grx3 (*SpGrx3*) was obtained from the mud crab *Scylla paramamosain*, which contains a 129-bp 5′ untranslated region, a 981-bp open reading frame, and a 1,183-bp 3′ untranslated region. The putative SpGrx3 protein contains an N-terminal thioredoxin domain and two C-terminal Grx domains. *SpGrx3* was expressed in all tissues examined, with the highest expression in the anterior gills. After hypoxia, *SpGrx3* expression was significantly up-regulated in the anterior gills of mud crabs. The expression of Grx2 and glutathione S-transferases was decreased, while the expression of glutathione peroxidases was increased following hypoxia when *SpGrx3* was silenced *in vivo*. In addition, the total antioxidant capacity of *SpGrx3*-interfered mud crabs was significantly decreased, and the malondialdehyde content was significantly increased during hypoxia. The subcellular localization data indicated that SpGrx3 was predominantly localized in the nucleus when expressed in *Drosophila* Schneider 2 (S2) cells. Moreover, overexpression of SpGrx3 reduced the content of reactive oxygen species in S2 cells during hypoxia. To further investigate the transactivation mechanism of *SpGrx3* during hypoxia, the promoter region of the *SpGrx3* was obtained by Genome Walking and three hypoxia response elements (HREs) were predicted. Dual-luciferase reporter assay results demonstrated that *SpGrx3* was likely involved in the hypoxia-inducible factor-1 (HIF-1) pathway during hypoxia, which could be mediated through HREs. The results indicated that *SpGrx3* is involved in regulating the antioxidant system of mud crabs and plays a critical role in the response to hypoxia.

## 1. Introduction

Hypoxia, in which the level of dissolved oxygen (DO) becomes ≤ 2 mg/L, frequently occurs and persists under intensive culture conditions [1]. Hypoxia can affect the behavior, physiology, and immunity of aquaculture species, resulting in low growth rates and high mortalities [2]. Numerous species of aquatic organisms, including the farmed marine species of mud crab, *Scylla paramamosain*, which is widely dispersed throughout China’s southeast coast, are susceptible to the effects of hypoxia [3]. As mud crabs widely inhabit intertidal zones, swamps, wetlands, and other changing environments; and often hide in silt or caves, they are vulnerable to hypoxic stress. However, little research has focused on how mud crabs cope with hypoxia.

Hypoxia can induce the formation of reactive oxygen species (ROS) in different organs, resulting in oxidative stress and eventually cell death [4,5]. Cells have antioxidant networks to scavenge excessive ROS [6]. The Glutaredoxin (Grx) redox system is a critical antioxidant system that participates in numerous physiological and biochemical processes, and is essential to the cell’s defenses against oxidative stress [7]. Grxs are small, conserved thiol-containing proteins belonging to the thioredoxin (Trx) superfamily [7]. Without exerting direct antioxidant characteristics, Grxs have the capacity to catalyze glutathione-dependent redox control to maintain and regulate the cellular redox state and redox-dependent signaling pathways [8,9]. Based on phylogenetic, sequence, and domain structure analyses, three distinct but functionally connected Grxs categories have been identified. The first category consists of the classical Grxs with two cysteines in their active site, exemplified by *Escherichia coli* Grx1 [10]. The second category, represented by *E. coli* Grx2, is a 24.3 kDa protein with three-dimensional structural similarities to the glutathione S-transferase (GST) family of proteins [11]. The third category includes the monothiol Grxs such as the yeast Grx3, Grx4, and Grx5 proteins. The monothiol Grxs can be further classified into two types: one with a single Grx domain and another with multiple Grx domains that include an N-terminal Trx domain and one to three C-terminal monothiol Grx domains [12]. All of the monothiol Grxs contain C-G-F-S as the active site motif [13]. To date, Grxs have been identified in bacteria [14], yeast [15], and mammals [16,17]. In aquatic organisms, Grxs have a significant influence on embryogenesis, brain and heart development, innate immunity, and environmental stress tolerance [18,19,20,21,22].

Grxs have also been found to play critical functions during hypoxia in previous studies [23,24,25,26]. For example, mammalian cells overexpressing Grxs proteins exhibited higher survival and proliferation rates, and lower oxidative damage following hypoxia-reoxygenation treatment [23]. Similarly, hypoxia-reoxygenation treatment significantly increased the expression of Grxs in cardiomyocytes and protected cardiomyocytes from apoptosis, oxidative stress, and inflammation induced by hypoxia-reoxygenation [24]. As hypoxia-inducible factor-1α (HIF-1α) is a key transcription factor regulating cellular oxygen homeostasis by activating genes involved in redox homeostasis [27,28], we hypothesized that a series of adaptive events of Grxs during hypoxia are also regulated by HIF-1α. Thus, in this study, a Grx3 gene (*SpGrx3*) was identified and cloned from *S. paramamosain*, and its mRNA expression patterns and potential functions were examined during hypoxia. The relationship between HIF-1α and *SpGrx3* was also investigated.

## 2. Materials and Methods

### 2.1. Animals and Hypoxia

Mud crabs (15 ± 3 g) were purchased from a crab farm in Jiangmen, Guangdong Province, China. Before experiments, mud crabs were acclimated in tanks for one week. The water temperature was 25 ± 1 °C, salinity was 10‰, and dissolved oxygen was 6.0 ± 0.2 mg/L. The water was changed every day. Mud crabs were fed oyster meat twice daily until 24 h before experimentation.

Mud crabs were divided into two groups, including the control group (dissolved oxygen: 6.0 ± 0.2 mg/L) and the hypoxia group (dissolved oxygen: 1.0 ± 0.2 mg/L) [3]. The hypoxic treatment was achieved within 10 min through the injection of nitrogen gas into the water and measured using a dissolved oxygen meter (JPB-607A, Shanghai, China). Each group was set up in triplicate. For tissue distribution analyses, the anterior gills, muscle, heart, stomach, hepatopancreas, hemocytes, and intestines were sampled and pooled from nine healthy *S. paramamosain*. For hypoxia experiments, the anterior gills were sampled at 0, 3, 6, 12, 24, and 48 h post-treatment, with each sample pooled from three crabs. All samples were immediately frozen in liquid nitrogen and stored at −80 °C until RNA extraction.

### 2.2. Gene Cloning and Sequence Analysis

Total RNA was extracted using a Trizol reagent (Invitrogen, Carlsbad, CA, USA) according to the manufacturer’s protocol. The concentrations of RNA were measured using a NanoDrop 2000 spectrophotometer (NanoDrop Technologies, Wilmington, DE, USA), and RNA integrity was verified by agarose electrophoresis. First-strand cDNA was synthesized from total RNA using the PrimeScript RT reagent Kit with gDNA Eraser (TaKaRa, Dalian, China), following the manufacturer’s instructions. A middle fragment of the *SpGrx3* cDNA sequence was initially obtained from our existing *S. paramamosain* transcriptome data [29]. Rapid amplification of cDNA ends as performed using the SMART RACE cDNA Amplification Kit (Clontech, CA, USA) following the manufacturer’s instructions. The forward and reverse primers were designed based on the partial *SpGrx3* cDNA sequence, and the PCR parameters were the same as those described previously [30].

A Genome Walking kit (TaKaRa, Dalian) was used to obtain the promoter region of *SpGrx3*. Genomic DNA from the gills of the mud crabs was prepared using the genomic DNA purification kit (Sangon, Shanghai, China) according to the manufacturer’s protocol. Three primers specific for *SpGrx3* (GSP-R1, GSP-R2, and GSP-R3) were designed according to the sequence of the 5′ region of the *SpGrx3* cDNA. In the first round of PCR, 100 ng of genomic DNA was used as the template, and GSP-R1 and AP1 (provided in the kit) were used as primers. In the second round of PCR, the amplification product from the first round of amplification was used as the template, and GSP-R2 and AP1 were used as the primers. The procedure was then repeated using primers GSP-R3 and AP1 in the third round of PCR. The amplification parameters of the three PCRs were carried out following the manufacturer’s instructions. All PCR products were then cloned into the pMD-18T vector (TaKaRa, Dalian) and sequenced.

The open reading frame (ORF) of *SpGrx3* was obtained using the ORF finder website (https://www.ncbi.nlm.nih.gov/orffinder/ (accessed on 12 January 2022)). The expert protein analysis system (https://web.expasy.org/compute_pi/ (accessed on 12 January 2022)) was used to analyze the deduced amino acid sequence and to predict the molecular weight (kDa) and isoelectric point (pI). The protein domain features were predicted by the Simple Modular Architecture Reach Tool (SMART; http://smart.embl-heidelberg.de/ (accessed on 12 January 2022)). Preliminarily protein three-dimensional structures were predicted by Phyre2 online software (http://www.sbg.bio.ic.ac.uk/phyre2 (accessed on 12 January 2022)) and transcription factor binding sites were predicted using JASPAR online software (https://jaspar.genereg.net/ (accessed on 12 January 2022)). Furthermore, a phylogenetic tree was constructed using the neighbor-joining method in MEGA 6.0 software (Mega Limited, Auckland, New Zealand).

### 2.3. RNA Interference Assay

The double-stranded RNAs (dsRNAs) of *SpGrx3* and GFP were produced using the T7 RiboMAXTM Express RNAi System (Promega, Madison, WI, USA) according to the manufacturer’s protocol. In dsRNA-mediated RNA interference experiments, mud crabs were injected with *SpGrx3* dsRNA (1 μg/g mud crab), while the control mud crabs were injected with equivalent quantities of GFP dsRNA. The RNAi efficiency was confirmed by detecting the expression of *SpGrx3* from anterior gill cDNA via real-time PCR at 48 h after dsRNA injection. To investigate the effects of hypoxia on *SpGrx3*-interfered mud crabs, mud crabs were exposed to hypoxia (dissolved oxygen 1.0 ± 0.2 mg/L) at 48 h post dsRNA injection. The anterior gill tissues were collected at 0, 3, 6, 12, 24, and 48 h after the induction of hypoxia, immediately frozen in liquid nitrogen, and then stored at −80 °C.

The relative expression levels of *SpGrx3* and several other antioxidant genes encoding GST, glutathione peroxidase (GPx), and Grx2 in the gills of *SpGrx3*-interfered mud crabs were detected by real-time PCR after hypoxia. 18S rRNA served as the internal control for the normalization of cDNA templates. Moreover, the gill tissues of *SpGrx3*-interfered mud crabs were used to detect the total antioxidant capacity (T-AOC) and the malondialdehyde (MDA) content after hypoxia using commercially available kits (Nanjing Jiancheng Chemical Industries, Nanjing, China). Briefly, the detection of T-AOC was achieved by an enzymatic reaction involving the deoxidization of ferric iron to ferrous iron [31]. MDA content was determined using the thiobarbituric acid reactive substances method [32]. All tests were performed in triplicate.

### 2.4. Real-Time PCR

The mRNA levels of *SpGrx3* in different tissues, and the expression pattern of *SpGrx3* mRNA during hypoxia were detected using real-time PCR. Real-time PCR was conducted using the SYBR Premix Ex Taq II (TaKaRa, Dalian) on a qTOWER^3^ 84 G Real-Time PCR Thermal Cycler (Analytik Jena, Jena, Germany). The thermal cycling parameters were 94 °C for 3 min to activate the polymerase, followed by 40 cycles of 95 °C for 15 s, 60 °C for 15 s, and 72 °C for 20 s. Each test was performed three times. 18S rRNA was used as the internal control for relative quantification. The data were analyzed using the 2^−ΔΔCt^ method [33].

### 2.5. Subcellular Localization of SpGrx3

*Drosophila* Schneider 2 (S2) cells were cultured at 28 °C in Schneider’s *Drosophila* media (Gibco, Carlsbad, CA, USA) supplemented with 10% fetal bovine serum (Gibco). For subcellular localization analyses, the ORF sequence of *SpGrx3* was amplified and inserted into the EcoRI/XhoI sites of pAc5.1-GFP vectors. S2 cells were transfected with the pAc5.1-*SpGrx3*-GFP recombinant vector or pAc5.1-GFP vector using FuGENE Transfection Reagent (Promega, Madison, Wisconsin, USA) according to the manufacturer’s instructions. After transfection for 48 h, the nuclei of S2 cells were stained with Hoechst 33,342 (Beyotime, Shanghai, China) and cells were then observed under a confocal laser scanning microscope (Leica TCS-SP5, Wetzlar, Germany).

### 2.6. Overexpression Assay

The ORF sequence of *SpGrx3,* without the stop codon, was amplified and inserted into the EcoRI/XhoI sites of pAc5.1-V5 vectors. Then, the pAc5.1-*SpGrx3*-V5 vector was transfected into S2 cells. The empty pAc5.1-V5 vector was transfected into S2 cells as the control. After transfection for 24 h, the cells were incubated in a hypoxic chamber (1% O_2_) for 24 h. The total intracellular ROS was determined by staining cells with 2,7-dichlorodihydrofluorescein diacetate (DCFH-DA). Briefly, cells were washed with phosphate buffer saline (PBS) and incubated with 5 μM DCFH-DA for 15 min at 37 °C. Cells were then washed three times with PBS and analyzed using a Spark^®^ Multimode Microplate Reader (TECAN, Switzerland).

### 2.7. Dual-Luciferase Reporter Assays

The ORF sequence of *SpHIF-1α* (GenBank accession number: KU644140.1) was amplified and inserted into the EcoRI/XhoI sites of pAc5.1-V5 vectors. A series of truncated *SpGrx3* promoters containing different numbers of hypoxia response elements (HREs) were generated by PCR and subcloned into the XhoI/HindIII site of the pGL3-Basic firefly luciferase reporter vector (Promega, Madison, Wisconsin). S2 cells were plated in a 24-well plate and transfected with the *SpGrx3* promoter activity reporter plasmid, the *SpHIF-1α* gene expression vector, and the pRL-TK Renilla luciferase plasmid (as internal control). Cells were harvested at 48 h post-transfection and lysed for the examination of firefly and Renilla luciferase activities using a Dual-Luciferase Reporter Assay System (Promega). All experiments were performed in triplicate.

### 2.8. Statistical Analyses

The data (mean ± SD) were analyzed using a one-way analysis of variance followed by Duncan’s multiple range test in SPSS 18.0 software (SPSS, Chicago, IL, USA). Differences were considered significant at *p* < 0.05.

## 3. Results

### 3.1. cDNA Cloning and Characterization of SpGrx3

The complete nucleotide and deduced amino acid sequence of *SpGrx3* are shown in Figure 1. The full-length *SpGrx3* contains a 129-bp 5′ untranslated region, a 1,183-bp 3′ untranslated region, and a 981-bp ORF encoding a 326 amino acids protein with a predicted molecular weight of 36.17 kDa and a predicted isoelectric point (pI) of 5.12 (GenBank accession number: OK639196). Two active sites of the monothiol Grx C-G-F-S (151–154 and 253–256 aa) and three conserved domains (an N-terminal thioredoxin domain and two C-terminal glutaredoxin domains) were predicted (Figure 2A), indicating that *SpGrx3* is a member of the Grx family [7].

The three-dimensional structure of the SpGrx3 protein is shown in Figure 2B, in which α-helix, β-sheet, and loop secondary structures were observed. As shown in Figure 3A, the amino acid sequences of *SpGrx3* were aligned with Grx3 of other species. The results showed that the amino acid sequences of Grx3 from the selected species were similar, including two C-terminal glutaredoxin domains and two highly conserved active sites (C-G-F-S). SpGrx3 protein has a high degree of homology with Grx3 from other crustaceans, including 96.32%, 80.06%, 79.45%, 78.53%, and 77.91% similarity to Grx3 proteins from *Portunus trituberculatus*, *Procambarus clarkii*, *Penaeus japonicus*, *Penaeus monodon*, and *Penaeus chinensis*, respectively (Figure 3A). In addition, the SpGrx3 protein exhibited 59.02% similarity to the *Danio rerio* Grx3 protein, 58.93% similarity to the *Homo sapiens* Grx3 protein, and 68.62% similarity to the *Mus musculus* protein (Figure 3A). A neighbor-joining phylogenetic tree was also constructed using the *SpGrx3* and Grx genes from different species (Figure 3B). The result revealed that Grx2, Grx3, and Grx5 were divided into three major clusters, with *SpGrx3* being most closely related to the Grx3 of *Portunus trituberculatus*.

### 3.2. Expression Profiles of SpGrx3

*SpGrx3* mRNA transcripts were detected in all of the tissues examined from *S. paramamosain*, with the highest expression in the gills, followed by that in muscle. However, expression was poor in hemocytes and the intestines (Figure 4A). The gills are important tissues in mud crabs as they are in direct contact with the external environment. Thus, we further investigated the expression of SpGrx3 in gills following hypoxia. As shown in Figure 4B, the expression of *SpGrx3* in the gills increased significantly from 6 to 48 h following hypoxia. The peak expression of *SpGrx3* in the gills was observed after 48 h of hypoxia, and was 8.73 times higher than that in the control group.

### 3.3. Expression Patterns of Antioxidant Genes in SpGrx3-Interfered Mud Crabs following Hypoxia

The efficiency of *SpGrx3* silencing was evaluated by real-time PCR. The results indicated that the expression of *SpGrx3* in the gills was significantly suppressed 48 h after injection of ds*SpGrx3* (Figure 5A). *SpGrx3* expression in control mud crabs injected with the same amount of GFP dsRNA was not significantly different.

As shown in Figure 5B, the relative expression of *SpGrx3* in ds*SpGrx3*-injected mud crabs was significantly inhibited after 48 h. Additionally, in dsGFP-injected mud crabs, significant increases in *SpGrx3* expression were still observed after the induction of hypoxia. From 0 to 48 h after hypoxia induction, the mRNA levels of *SpGrx3* in *SpGrx3*-interfered mud crabs were lower than those in dsGFP-injected mud crabs. The relative expression of Grx2 is shown in Figure 5C. The *Grx2* mRNA levels in ds*SpGrx3*-injected mud crabs were increased after hypoxia and peaked at 6 h. Compared to dsGFP-injected mud crabs, the transcription levels of *Grx2* in *SpGrx3*-interfered mud crabs were significantly higher at 0 h (the 48 h of dsRNA injection), but significantly lower at 3, 6, and 12 h after hypoxia. Similarly, the relative expression level of *GST* in *SpGrx3*-interfered mud crabs was significantly higher than that in the dsGFP group at 0 h (48 h after dsRNA injection), but significantly lower than that in the dsGFP group at 3 and 12 h after hypoxia induction (Figure 5D). The relative level of *GPx* is shown in Figure 5E. Compared to dsGFP-injected mud crabs, *GPx* mRNA expression was significantly higher at 0, 3, 6, and 48 h (*p* < 0.05) in ds*SpGrx3*-injected mud crabs after hypoxia.

### 3.4. T-AOC and MDA Contents in SpGrx3-Interfered Mud Crabs after Hypoxia

As shown in Figure 6A, T-AOC increased rapidly at 3 h after hypoxia then decreased slightly from 6 to 24 h, and reached the highest level at 48 h in both the dsGFP group and the ds*SpGrx3* group. However, T-AOC in the ds*SpGrx3* group was significantly decreased at 12 h and 48 h compared to those in the dsGFP group. As shown in Figure 6B, the levels of MDA increased from 3 to 48 h after hypoxia in both the dsGFP group and the ds*SpGrx3* group. A significant increase in MDA content was found at 6, 12, and 48 h in the ds*SpGrx3* group compared to the dsGFP group.

### 3.5. The Subcellular Localization of SpGrx3

The subcellular localization of SpGrx3 in S2 cells was observed using confocal laser scanning microscopy. After pAc5.1-*SpGrx3*-GFP transfection into S2 cells for 48 h, the localization of pAc5.1-*SpGrx3*-GFP was dispersed throughout the nucleus, suggesting that SpGrx3 protein was localized in the nucleus of S2 cells (Figure 7).

### 3.6. Effect of SpGrx3 Overexpression on Intracellular ROS Content after Hypoxia

To investigate the function of SpGrx3, S2 cells transfected with the pAc5.1-*SpGrx3* expression vector or the empty pAc5.1-V5 vector were incubated under hypoxic conditions for 24 h, after which, the ROS levels in S2 cells were measured. The results showed that the overexpression of SpGrx3 reduced the ROS levels in S2 cells after hypoxia (Figure 8).

### 3.7. Analysis of the SpGrx3 Promoter

The 1527-bp *SpGrx3* promoter was predicted to contain three hypoxia response elements (HREs, 5′-RCGTG-3′, the binding motif of HIF-1α), suggesting that the expression of *SpGrx3* may be regulated by HIF-1α (Figure 9A). Thus, the effects of *S. paramamosain* HIF-1α on the *SpGrx3* promoter fragments were analyzed using dual-luciferase reporter assays (Figure 9B). The luciferase activity of the full-length *SpGrx3* promoter (pGL3-*SpGrx3*-F1) was significantly increased by 6.42-fold compared to that of the empty vector (pGL3-Basic), while the deletion of one HRE (pGL3-*SpGrx3*-F2) resulted in a 10.34-fold increase in luciferase activity and the deletion of two HREs (pGL3-*SpGrx3*-F3) resulted in a 6.77-fold increase in luciferase activity. Furthermore, the luciferase activity of the pGL3-*SpGrx3*-F3.1 vector (the pGL3-*SpGrx3*-F3 vector mutant with HRE mutations of 5′-CTGCGTGA-3′ to 5′-CTATACAA-3′) was significantly decreased. Those data indicated that *SpGrx3* might be a target of HIF-1α and that HIF-1α may activate the transcription of *SpGrx3* via HRE.

## 4. Discussion

In this study, the full-length cDNA of Grx3 was first identified and cloned from *S. paramamosain*. From the predicted amino acid sequence, a Trx domain and two typical Grx structural domains with the active center sequence C-G-F-S were identified, suggesting that *SpGrx3* is a monothiol Grx [12]. Multiple comparisons of SpGrx3 with the Grx3 from other species suggested that all of the analyzed Grx3 share a common basic active site, C-G-F-S, which is highly conserved across species. Phylogenetic analyses revealed that *SpGrx3* was clustered with other crustaceans, indicating that Grx3 is highly conserved across closely related species. We hypothesize that the conserved Grx structural domain plays similar roles in various aspects of crustacean physiology and biochemistry.

The tissue distribution of Grx3 has been studied in some aquatic species. For example, Grx3 is highly expressed in the zebrafish heart, brain, eye, and ventral region [34]. Similarly, Grx3 is also widely distributed throughout different crustacean tissues. It has also been reported that Grx3 has the highest expression level in the muscle of black tiger shrimp (*Penaeus monodon*) [18] and the intestinal tissue of white shrimp (*Litopenaeus vannamei*) [35]. In the present study, the expression of *SpGrx3* was detected in all tissues examined, but varied greatly across different tissues. Thus, those data suggest that *SpGrx3* has a wide range of biological functions in different organs. The highest *SpGrx3* expression was found in the anterior gills, which are in direct contact with the external environment and have the ability to control the deleterious effects of ROS produced by environmental stresses [36]. As SpGrx3 plays an important role in maintaining tissue redox homeostasis, we speculated that the high levels of *SpGrx3* expression in the gills may be a strategy to prevent oxidative damage. Thus, we further investigated the expression of *SpGrx3* in the anterior gills after hypoxia. Hypoxia is one of the most common aquatic environmental stressors, and can cause the accumulation of ROS in cells [37,38]. The antioxidant defense system is the first line of defense against oxidative stress. As a part of the ROS-scavenging system, Grx3 can be induced by stress to maintain and regulate the cellular redox state [39]. Fan et al. [18] found that the expression of Grx3 was induced by ammonia-N stress and bacterial infections in the hepatopancreas and gills of *Penaeus monodon*. Similarly, the expression of Grx3 in the hepatopancreas and gills of *Litopenaeus vannamei* was significantly upregulated after ammonia-N stress and lipopolysaccharide injection [35]. In the current study, the transcription of *SpGrx3* was significantly upregulated in gills after hypoxia, which suggested that *SpGrx3* plays an important role in the antioxidant defense system of mud crabs during hypoxia.

To further validate the defense mechanism of *SpGrx3* in mud crabs coping with hypoxia, we inhibited the expression of *SpGrx3* using RNAi. The expression of *Grx2*, *GST*, and *GPx*, which evolved from a common thioredoxin-like ancestor [40], was analyzed after *SpGrx3* expression was inhibited. The results showed that *SpGrx3*, *Grx2*, and *GST* had lower expression levels in *SpGrx3*-interfered mud crabs, whereas a significant increase in *GPx* mRNA expression was observed. The *GPx* family is a crucial part of the cellular detoxification system, as it catalyzes the conversion of lipid hydroperoxide and hydrogen peroxide into water and alcohol [41]. Since *Grx* and *GPx* share a common thioredoxin fold and have high affinity for glutathione, it is possible to engineer *Grx* into *GPx* and vice versa [42]. These results show that *SpGrx3* expression was closely linked to the expression of other antioxidant genes and that *GPx* appears to partially compensate for *SpGrx3* function in response to hypoxia. Furthermore, compared to the control group, *SpGrx3*-interfered mud crabs had considerably lower T-AOC and significantly higher MDA levels following hypoxia. A similar result has also been reported following the knockdown of *Grx3* expression in *Litopenaeus vannamei*, which aggravated oxidative damage to proteins and lipids [35]. The decrease in such antioxidant capabilities can lead to oxidative damage. Polyunsaturated fatty acids are damaged by oxidative stress, resulting in lipid peroxidation. MDA is a common indicator of lipid peroxidation [32]. Our results suggested that the inhibition of *SpGrx3* led to a greater risk of oxidative damage in mud crabs.

In general, the subcellular localization of Grxs is correlated with its different functions in the cell [43]. For example, nuclear Grx3 may prevent damage to transcriptional mechanisms and regulate gene expression in response to oxidative stress [44]. Due to the fact that *S. paramamosain* has not yet established mature and stable cell lines, in this study, we used the S2 cell line of *Drosophila*, which the evolutionary position is closer to *S. paramamosain*, to study the subcellular localization of SpGrx3. In the present study, the SpGrx3-GFP fusion protein was observed in the nucleus of transfected S2 cells, suggesting that SpGrx3 may play an important role in the nucleus. Within the Grx family, Grx3, Grx4, and Grx5 are monothiol Grxs, but have different subcellular localizations. Grx5 is located in the mitochondrial matrix [45], while Grx3 and Grx4 are both located in the nucleus [46,47]. The differences in subcellular localization may be mediated by the unique Trx domains in Grx3 and Grx4 [47]. A previous study has also shown that the nuclear accumulation of mammalian Grx3 occurs only under oxidizing conditions and that such nuclear accumulation is reversible [48]. Therefore, the nuclear localization of SpGrx3 may be affected by multiple factors, which need to be further studied. In addition, our results also showed that S2 cells overexpressing SpGrx3 had lower ROS levels after hypoxia, thus, supporting the role of SpGrx3 as a ROS scavenger. A previous study demonstrated that overexpression of Grx can protect cells from oxidative damage induced by hydrogen peroxide [49]. Compared to the controls, overexpression of Grx proteins in HEK293 and HeLa cells under hypoxic and reoxygenation conditions exhibited higher survival and proliferation rates, and lower oxidative damage [23]. Our results suggest that SpGrx3 is involved in the regulation of the antioxidant system and plays a critical role in the response to hypoxia.

The mechanism by which hypoxia induces increases in the expression of *SpGrx3* was elucidated in this study. HIF-1α is one of the most important transcription factors that allow organisms to tolerate hypoxia. Under hypoxic conditions, HIF-1α can bind to E-box-like HREs present in the promoter regions of downstream genes, and activate their transcription [50,51,52]. HRE is an important regulatory sequence mediating the hypoxic responses of cells. Mutations to HRE sites will cause the genes regulated by HIF-1α to lose their transcriptional response to hypoxia [53]. In the present study, three HREs were found in the promoter region of *SpGrx3*, indicating that *SpGrx3* may be a downstream effector gene regulated by HIF-1α. Moreover, the results of dual-luciferase reporter assays suggested that the hypoxia-dependent activation of the *SpGrx3* promoter involves the HRE motif and HIF-1α, which was consistent with similar findings in mammals [54,55,56]. However, the luciferase activity driven by the full-length *SpGrx3* promoter was lower than that of the *SpGrx3* promoter fragment containing fewer HREs, which was different from the findings of previous studies, which indicated that the loss of HREs reduced the transcriptional activity of the promoter [57,58]. Nonfunctional core HREs have been found in other genes [59,60], which prompted us to address the possibility that the *SpGrx3* promoter contains a negative transcriptional region between the -1108 to -184 sites (the deleted region of pGL3-*SpGrx3*-F2 relative to the full-length *SpGrx3* promoter). Therefore, the response mechanism of *SpGrx3* to hypoxia may be complex, which deserves further study.

## 5. Conclusions

In this study, the cDNA of *SpGrx3* was cloned and found to belong to the Grxs family. The *SpGrx3* gene was observed in all tissues examined, with the highest expression in the anterior gills. The expression of *SpGrx3* was increased following hypoxia and was closely associated with the expression of other antioxidant genes. The suppression of *SpGrx3* expression reduced T-AOC and resulted in an increase in oxidative damage following hypoxia, while the overexpression of *SpGrx3* in S2 cells reduced the ROS levels following hypoxia. Furthermore, *SpGrx3* is likely regulated by HIF-1α through HREs during hypoxia.

## Figures and Tables

**Figure 1 antioxidants-12-00076-f001:**
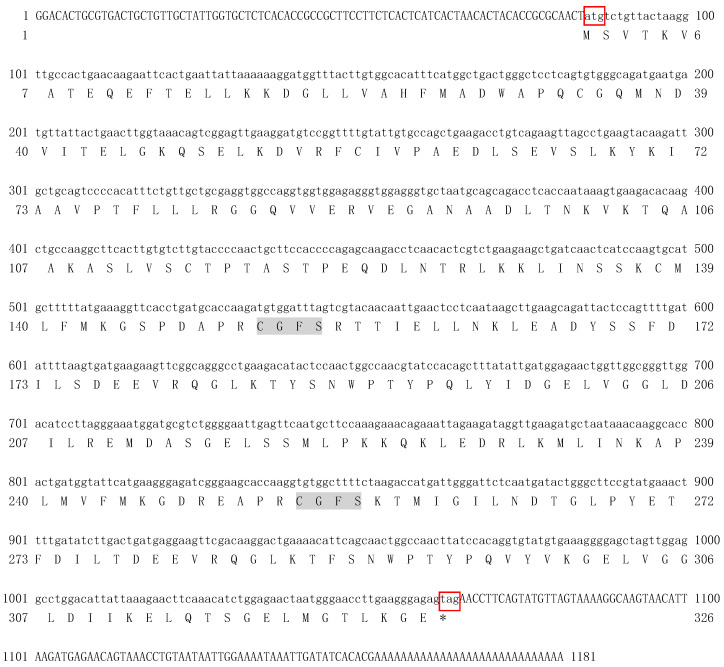
Nucleotide and deduced amino acid sequences of *SpGrx3*. The letters in the red box indicated the start codon (atg) and stop codon (tga). The shading indicates the catalytic amino acids (C-G-F-S).

**Figure 2 antioxidants-12-00076-f002:**
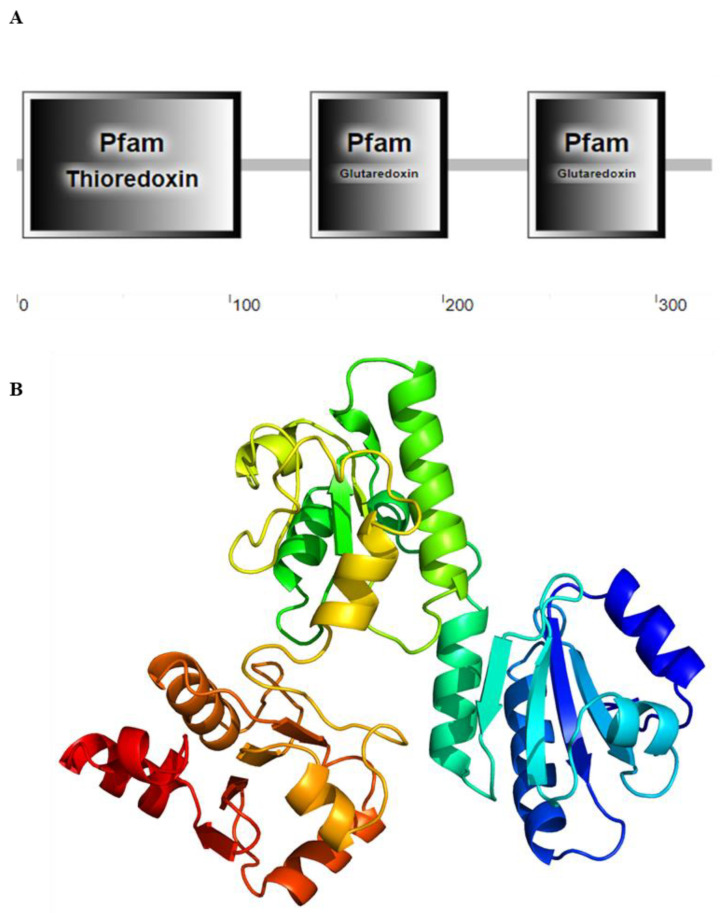
The deduced domain and three-dimensional structures of the SpGrx3 protein. (**A**) The deduced domain of SpGrx3 was predicted using SMART. An N-terminal thioredoxin domain and two C-terminal glutaredoxin domains were predicted. (**B**) The three-dimensional structures of the SpGrx3 protein were predicted by Phyre2 online software. The image was colored by the rainbow in the direction of the N to C termini.

**Figure 3 antioxidants-12-00076-f003:**
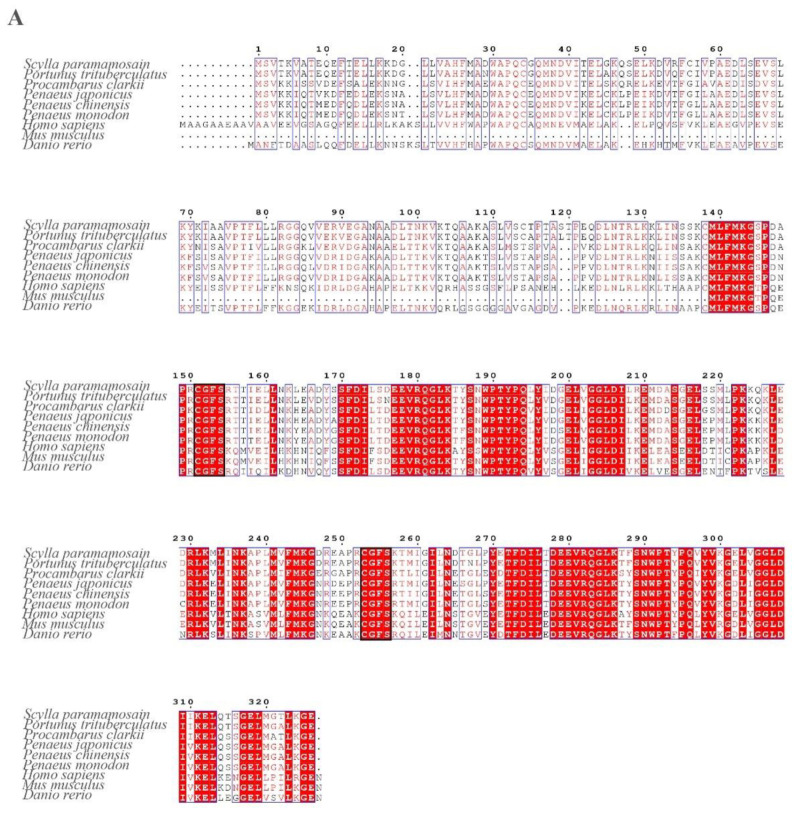
(**A**) Multiple sequence alignment of Grx3 from other species. Red regions indicate conserved amino acid residues. Red letters indicate similar residues. The conserved active sites, C-G-F-S, are shown in the black boxes. The results of the amino acid sequence counts are shown at the top of each row. (**B**) A neighbor-joining phylogram of *SpGrx3* with Grxs sequences from other species. *SpGrx3* is indicated by a triangle.

**Figure 4 antioxidants-12-00076-f004:**
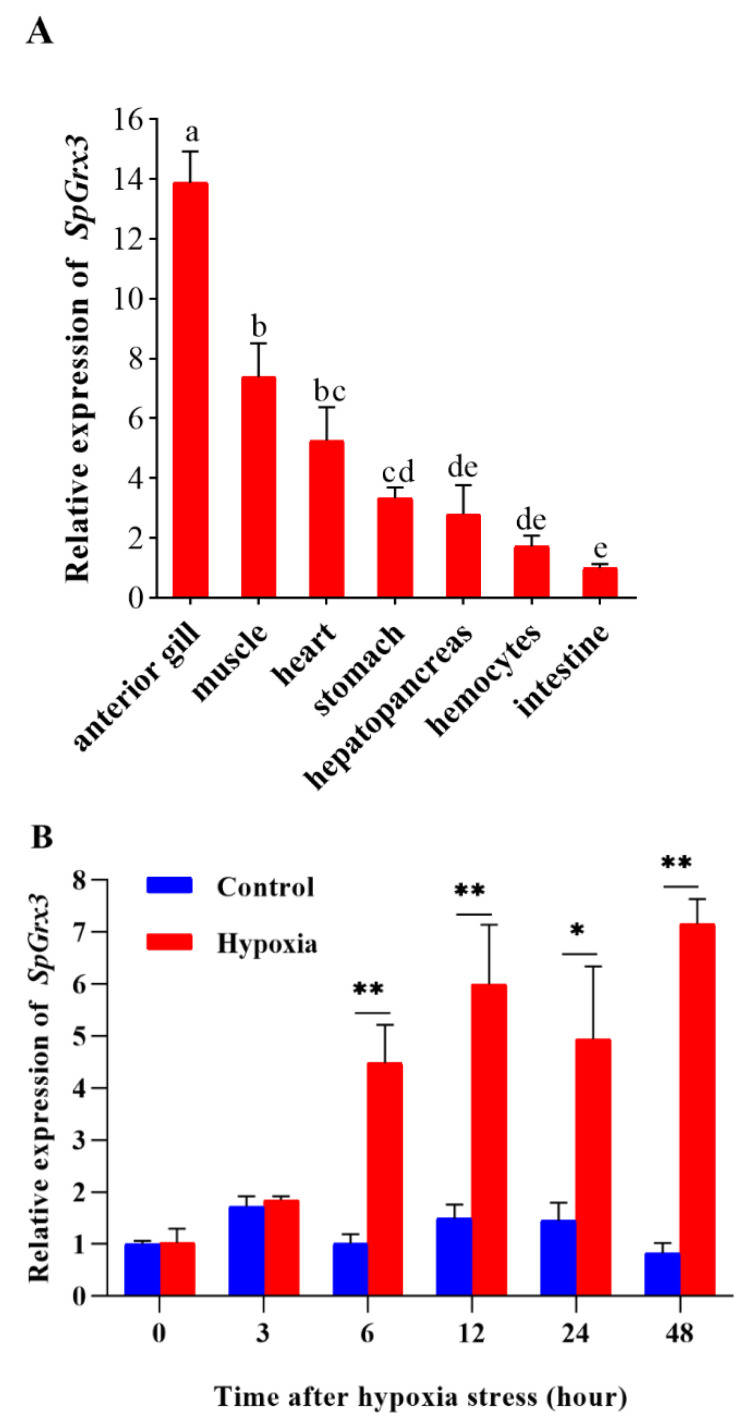
Relative expression levels of *SpGrx3*. (**A**) The mRNA level of *SpGrx3* in different tissues. Data are presented as the mean ± SD (N = 3). Different letters denote significant differences (*p* < 0.05). (**B**) The expression profiles of *SpGrx3* in gills after hypoxia. Data are presented as the mean ± SD (N = 3). *: *p* < 0.05, **: *p* < 0.01.

**Figure 5 antioxidants-12-00076-f005:**
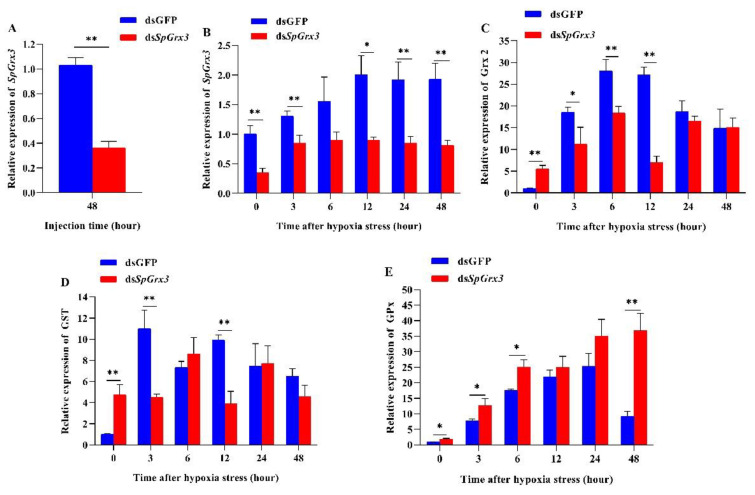
Effects of *SpGrx3* silencing on the expression of antioxidant-related genes. (**A**) Silencing efficiency of *SpGrx3* in the gills after dsRNA injection. Relative expression levels of *SpGrx3* (**B**), Grx2 (**C**), GST (**D**), and GPx (**E**) in the gills of *SpGrx3*-interfered mud crabs following hypoxia. Data are presented as the mean ± SD (N = 3). *: *p* < 0.05, **: *p* < 0.01.

**Figure 6 antioxidants-12-00076-f006:**
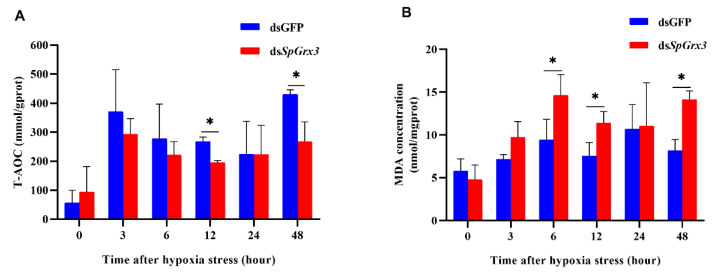
Functional analysis of *SpGrx3* during hypoxia. T-AOC (**A**) and MDA (**B**) concentrations in the gills of *SpGrx3*-interfered mud crabs after exposure to hypoxia (the dsGFP group was used as the control). The data are presented as the mean ± SD (N = 3). *: *p* < 0.05.

**Figure 7 antioxidants-12-00076-f007:**
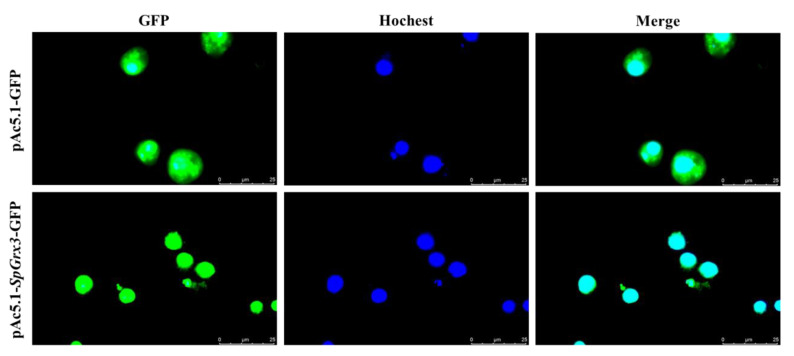
The subcellular localization of SpGrx3 protein in S2 cells. S2 cells were transfected with pAc5.1-GFP (upper row) or pAc5.1-*SpGrx3*-GFP (lower row). The cell nucleus was stained with Hoechst33342 (blue) and fluorescence was observed using confocal laser scanning microscopy.

**Figure 8 antioxidants-12-00076-f008:**
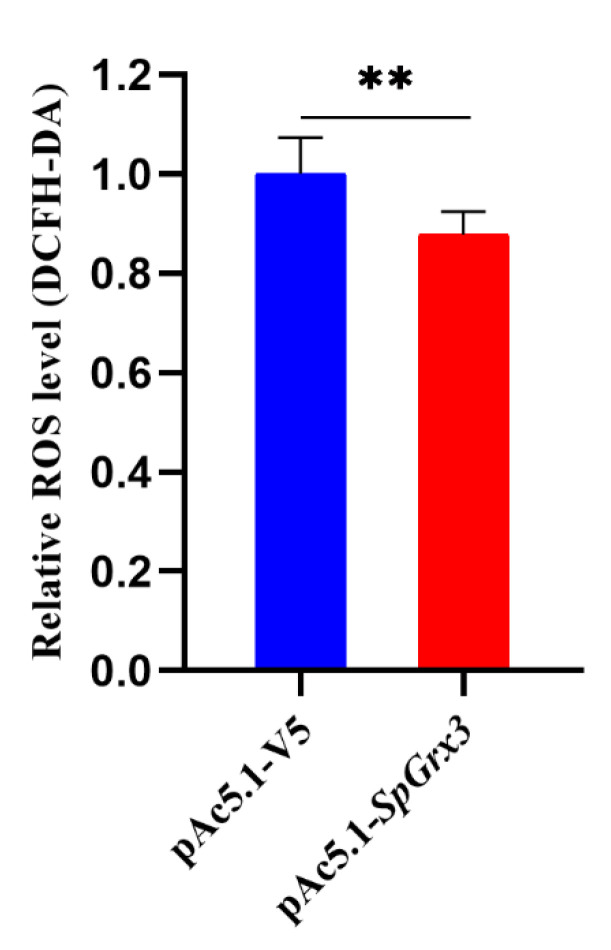
Relative ROS levels in S2 cells. After transferring the empty pAc5.1-V5 vector or the pAc5.1-*SpGrx3* expression plasmid into S2 cells for 24 h, cells were incubated in a hypoxic chamber (1% O2) for 24 h. The ROS levels were then detected by DCFH-DA staining as described in the methods. The data are presented as the mean ± SD (N = 3). **: *p* < 0.01.

**Figure 9 antioxidants-12-00076-f009:**
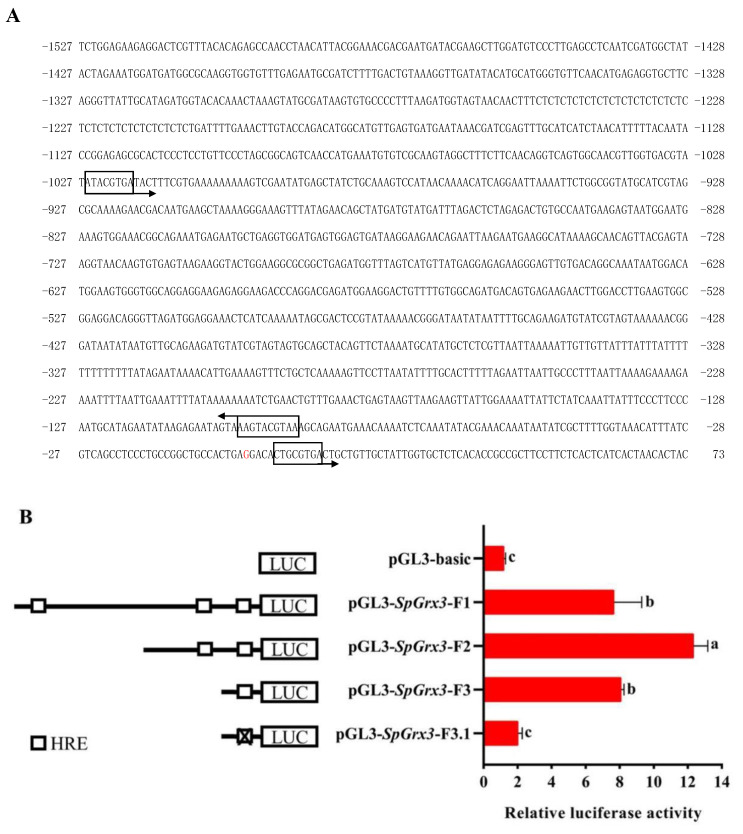
Putative promoter and HRE regulatory elements in the 5′ upstream sequence of *SpGrx3*. (**A**) Schematic diagram of the gene promoter region. The transcriptional start site is shown in red font. The predicted HRE is indicated by a black box. The arrow indicates the direction of the HRE motif. (**B**) The activity of each plasmid relative to empty vector (pGL3-Basic) in S2 cells. Different 5′ truncations of the *SpGrx3* promoter were individually cloned into the pGL3-Basic vector. The HRE site, CTGCGTGA, was mutated to CTATACAA in pGL3-*SpGrx3*-F3.1. S2 cells were co-transfected with the firefly luciferase reporter plasmid, expression plasmids (pAc5.1-*SpHIF-1α*-V5), and the control plasmid (pRL-TK Renilla luciferase plasmid). The data are expressed as fold induction relative to the empty vector (pGL3-Basic). The bars represent the mean ± SD (N = 3). *p* < 0.05.

## Data Availability

Data are contained within the article and Appendix A.

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
