# Peer review of "Hypoxia Affects the Antioxidant Activity of Glutaredoxin 3 in Scylla paramamosain through Hypoxia Response Elements"

_antioxidants, 2022, doi:10.3390/antiox12010076_

Round 1

Reviewer 1 Report

Mud crab Scylla paramamosain is widely distributed throughout China and Southeast Asian countries. As the mud crab widely inhabits the intertidal zone, swamp, wetland, and other complex and unstable environments, it is more vulnerable to hypoxia stress. However, little research has focused on how mud crab cope with hypoxia. Hypoxia may affect the animal behavior, physiology, and immunity of aquaculture species, resulting in low growth rates and high mortalities. 

Glutaredoxin (Grx) is a key member of the thioredoxin superfamily, which plays an important role in host defense against oxidative stress. At present, the role of Grx in the hypoxia response remains unclear for crustaceans. Therefore, this study was conducted to clone the full-length of Grx3 gene (SpGrx3) from Scylla paramamosain, and its mRNA expression and potential functions were further examined after hypoxia challenge. These results will provide important for understanding the function of SpGrx3 during the hypoxia response. My major comments are as the follows.

1) Fig 4A, Lines 80-85, and lines 156-157. The major functions of anterior gill and posterior gill have some differences, i.e. anterior gills are mainly responsible for respiration while posterior gill are more important for osmotic regulation for crustacean. It is better to sample the second or third pairs of gill for investigate the gene-expression and hypoxia response of this species. However, the authors did not mention the sampling detail in ‘2.1. Animals and hypoxia’. Please provide this information in ‘Materials and methods’, and try to say sth in this topic of the discussion.

Romano, N., Zeng, C., 2010. Survival, osmoregulation and ammonia-N excretion of blue swimmer crab, Portunus pelagicus, juveniles exposed to different ammonia-N and salinity combinations. Comparative Biochemistry and Physiology Part C: Toxicology and Pharmacology 151, 222-228.

Romano, N., Wu, X.G., Zeng, C.S., Genodepa, J., Ellman, J., 2014. Growth, osmoregulatory responses and changes to the lipid and fatty acid composition of organs from the mud crab, Scylla serrata, over abroad salinity range. Marine Biology Research 10, 460-471.

2) Lines 148-158. For RNA interference assay, why did the authors only have one injection during the 48h experiment, could you please provide some references or evidences in here.

3) Line 330, the discussion seems loose not focus on the important finding, could you discussion the potential mechanism and signal pathway of SpGrx3 involved in hypoxia challenge of Scylla paramamosain? If possible, the authors could provide the figure to illustrate the potential mechanism or signal pathway of SpGrx3 involved in hypoxia challenge, which are important to conclude your important finding and highlight.

Reviewer 2 Report

The manuscript by Jie and colleagues reports on the cloning, expression and functional analysis of glutaredoxin 3 gene from the mud crab. The study is comprehensive, well written and the conclusions are supported by the data shown. Most of the comments below relate to grammatical or stylistic issues.

1. L16 129-bp

2. The authors use the fully italicized name throughout, regardless of whether this is for the gene, the transcript or the protein. The protein should not be italicized. There are multiple instancesof this throughout the manuscript.

3. L20 nucleus when expressed in Drosophila

4. L27 investigate

5. L45 crabs

6. L52 What defines “tiny” rather than “small”?

7. L64 E. coli should be italicized

8. L64 This is misleading. It suggest only a few have been demonstrated. I would think that other bacteria have been tested and refs 18-22 provide yet other examples that are not listed.

9. L89 10 min (to match styles used predominately in the ms)

10. L90, L96, L97, L99, L106, L134, L141, L145, L146, L147, L177,    Please include the City and country for all companies, but only on first mention. For US companies, provide the company name, city, state and USA). Please double check throughout.

11. L95 the mud crab

12. L98, L99 NanoDrop

13. L101 vs L110 and elsewhere. Is it only the Japan-located company that uses the TaKaRa spelling?

14. L113 the kit

15. L127/232 phylogenic or phylogenetic?

16. L127 Please provide source or reference for MEGA

17. In the molecular methods sections, please make sure that Table S1 is clearly referred to

18. L147 Is this a confocal (L238)?

19. L150, L187 USA is not required here (given earlier)

20. L159 genes (encoding glutathione..

21. L175 Please define PBS on first use

22. L177 by a Spark

23. L181 define HRE on first use in main text

24. L195 are shown

25. L202 is shown

26. L209, L211, L212 Please use identity or similarity (whichever it is) for greater precision in meaning

27. L235 transcripts were detected

28. L240 S2 cells

29. L251, L309, L331, L399  After first use the genus names can be abbreviated

30. L271 (two instances), L273, L275, L278-9. These are genes/mRNAs that are being measured so should be italicized

31. L298, L441, L411 Although this may be significant, it is only a 10% reduction

32. L318 and L418  Perhaps use activate rather than initiate as it is RNA Pol that is carrying out the initiation

33. L321 in the

34. L323 Please make sure the arrows are shown (by printed copy did not have them)

35. L331 full length cDNA of …….  From S.

36. L353 As SpGrx3 has an important role in maintaining.. (it also should not be italicized wen referring to protein, also L359, L366 etc)

37. L359 nucleus in S2 cells

38. L376 upregulation

39. L382 Please provide a reference for this fact, also L387

40. L415 induces increased expression of SpGrx3 was elucidated

41. L416 study. HIF

42. L436 into the Grxs family. The Sp…

43. The journal titles use multiple different styles. Some are in full, some abbreviated, some capitalized, others not.

Reviewer 3 Report

The paper aims to illustrate, in the crustacean Scylla paramamosain, the relationship between hypoxia, HIF-1, and the antioxidants system, in particular, Glutaredoxin (Grx), a member of the thioredoxin superfamily. The purpose is to explore whether HIF-1 is involved in the regulation of a Grx3 gene (SpGrx3), and the potential functions of protein expression.  

Appropriate techniques are used. In general, the text if fluid but it suffers of typos and style errors that require a revision of an English speaker with experience in scientific writing. 

The paper is valuable for readers interested in the analysis of the pathways that modulate the response to hypoxia to different animal species. However, the value of the paper is decreased by several important flaws that prevent it is accepted for publication, formed as it is.    

A major concern is the confusion existing when considering the global approach and the rationale/advantage for choosing different biological systems (e.g. S2 cells and native crab extracts) for the different analyses. For example, it appears that many tissues have been used to analyse the tissue distribution of SpGrx3 but only hepatopancreas and gills have been used to describe the expression levels of SpGrx3 after hypoxia. At the same time, the hepatopancreas was not used for other analyses. Moreover, S2 cells were used to evaluate the subcellular localization of SpGrx3. But this cell line is not from crab. Why did authors decide to include S2 cells in the study? See for example, Results 3.2, concerning SpGrx3 tissue distribution and intracellular localization. In addition, since the effect of hypoxia is recognized to be animal-, tissue- and cell-specific, the rationale for using S2 cells must be clearly elucidated. In my opinion, the results on S2 cells can be only related to this cell type and do not provide indication that can be integrated with data on Grx3 in the mud crab. I would prefer, if possible, that the cell localization of SpGrx3 is analysed in native crab cells. Differently, data on S2 cells require proper discussion. 

Moreover, the emphasis given to the role of HIF-1 is not based on conclusive results but only on an indirect evidence provided by the presence of HREs in SpGrx3. This prevents any conclusion about a true correspondence between HIF-1 and Grx3 in the crab. This has to be properly clarified. 

Other points:

the number of samples should be increased for better statistic evaluation;

the oxidative status and the ROS production need to be evaluated on the same crab-derived biological substrates;

in Material and Methods, the paragraph 2.2. should be better organized;

please, provide appropriate reference for normoxia at DO= 6,00 mg/L and DO= 1 mg/L for hypoxia.

In general, the paper requires a critical revision to address the above aspects. 
